# Synthesis, Photophysical Characterization, and Sensor Activity of New 1,8-Naphthalimide Derivatives

**DOI:** 10.3390/s20143892

**Published:** 2020-07-13

**Authors:** Stanislava Yordanova-Tomova, Diana Cheshmedzhieva, Stanimir Stoyanov, Todor Dudev, Ivo Grabchev

**Affiliations:** 1Faculty of Chemistry and Pharmacy, Sofia University “St. Kliment Ohridski”, 1 J. Baurchier blvd., 1164 Sofia, Bulgaria; ohsy@chem.uni-sofia.bg (S.Y.-T.); dvalentinova@chem.uni-sofia.bg (D.C.); ohss@chem.uni-sofia.bg (S.S.); t.dudev@chem.uni-sofia.bg (T.D.); 2Faculty of Medicine, Sofia University “St. Kliment Ohridski”, 1 Koziak str., 1407 Sofia, Bulgaria

**Keywords:** 1,8-naphthalimides, photophysics, metal ions, pH

## Abstract

Three new 1,8-naphthalimide derivatives **M1**–**M3** with different substituents at the C-4 position have been synthesized and characterized. Their photophysical properties have been investigated in organic solvents of different polarity, and their fluorescence intensity was found to depend strongly on both the polarity of the solvents and the type of substituent at C-4. For compounds **M1** and **M2** having a tertiary amino group linked via an ethylene bridge to the chromophore system, high quantum yield was observed only in non-polar media, whereas for compound **M3**, the quantum efficiency did not depend on the medium polarity. The effect of different metal ions (Ag^+^, Ba^2+^, Cu^2+^, Co^2+^, Mg^2+^, Pb^2+^, Sr^2+^, Fe^3+^, and Sn^2+^) on the fluorescence emission of compounds **M1** and **M2** was investigated. A significant enhancement has been observed in the presence of Ag^+^, Pb^2+^, Sn^2+^, Co^2+^, Fe^3+^, as this effect is expressed more preferably in the case of **M2**. Both compounds have shown significant pH dependence, as the fluorescence intensity was low in alkaline medium and has been enhanced more than 20-fold in acidic medium. The metal ions and pH do not affect the fluorescence intensity of **M3**. Density-functional theory (DFT) and Time-dependent density-functional theory (TDDFT) quantum chemical calculations are employed in deciphering the intimate mechanism of sensor mechanism. The functional properties of **M1** and **M2** were compared with polyamidoamine (PAMAM) dendrimers of different generations modified with 1,8-naphthalimide.

## 1. Introduction

Fluorescence analysis of various analytes is a highly sensitive analytical method. It is based on the interaction between the analyte and a receptor fragment bound to a fluorescent molecule, resulting in changes in its photophysical characteristics [1]. Since the interaction is at the molecular level, extremely low concentrations of analytes can be detected by this method, making it highly effective. Fluorescence sensors operating on the principle of photoinduced electron transfer (PET) are of great scientific interest, because their emission can be selectively ‘switched on’ or ‘switched off’ in the presence of analytes [2,3,4]. Various fluorescence chromophore systems such as 1,8-naphthalimide, benzanthrone, xanthene derivatives, etc. are used as a signal fragment in the design of PET sensor systems [5,6,7,8,9,10,11,12,13]. Very important among these are the derivatives of 1,8-naphthalimide, which depending on the nature of the C-4 substituents, may emit blue or yellow-green fluorescence [14,15,16,17]. In most cases, this position contains receptor fragments that can react with the analytes tested, which determines the color of the fluorescence emitted by the sensor [4,18,19]. In the design of sensing systems for the detection of metal ions or protons, 1,8-naphthalimides can be used as a signaling fragment. Tertiary amines attached to the chromophore system via an ethylene spacer are mainly used as receptors in the detection of metal ions and protons. Such 1,8-naphthalimides can be used as individual molecules [20,21,22,23,24] or incorporated into dendrimer structures [25,26,27,28], hyperbranched polymers [29,30], or linear polymers [31,32,33,34]. In this way, the selectivity and sensitivity of the sensors can be varied. In our previous works, we have been used 1,8-naphthalimides having as a substituent at the C-4 position *N*,*N*-dimethylethylenediamine residue or a similar cyclic system such as *N*-methylpiperazine for the peripheral modification of PAMAM dendrimers from zero (with four 1,8-naphthalimides) first (with eight 1,8-naphthalimides) and third (with 16 1,8-naphthalimides) generations. It has been shown that the sensor capacity of these dendrimers depends strongly on their generation [5,35,36].

In recent years, Density-functional theory (DFT) and Time-dependent density-functional theory (TDDFT)methods have proven themselves as useful and reliable tools for designing and developing fluorescence sensors [37,38]. They provide an accurate picture of the electronic structure of fluorescent compounds, thus giving insight into the intimate mechanism of relevant photophysical processes.

The aim of this work is to synthesize three new derivatives of 1,8-naphthalimide with substituents at the C-4 atom, respectively *N*,*N*-dimethylethylendiamino, *N*-methylpiperazine, and *N*-propylamine. The substituent bonded to the imide nitrogen atom was *N*-Acetylethylenediamine. Those derivatives can be considered as a part of the internal PAMAM dendrimer structure. The functional properties of the new compounds thus obtained were investigated and compared with those of the corresponding dendrimers.

## 2. Materials and Methods

UV/Vis absorption spectra were recorded on a “Thermo Spectronic Unicam UV 500” spectrophotometer. Fluorescence investigations were performed on a “Cary Eclipse” spectrophotometer. In all spectroscopic measurements, 1 cm path length synthetic quartz glass cells were used. The organic solvents (spectroscopic grade) used in this study were: chloroform (CHCl_3_), ethyl acetate (EtOAc), dichloromethane (DCM), 1,4-dioxane (Dioxane), dimethyl sulfoxide (DMSO), acetonitrile (MeCN), *N*,*N*-dimethylformamide (DMF), methanol (MeOH), and ethanol (EtOH).

The fluorescence quantum yield has been calculated on the basis of the absorption and fluorescence spectra by Equation (1).
(1)$$\Phi_{F}=\Phi_{\mathrm{st}}\frac{S_{u}}{S_{\mathrm{st}}}\frac{A_{\mathrm{st}}}{A_{u}}\frac{n_{\mathrm{Du}}{}^{2}}{n_{\mathrm{Dst}}{}^{2}}$$
where the Φ_F_ is the emission quantum yield of the sample; Φ_st_ = 0.78 in ethanol [39] is the emission quantum yield of the standard (Coumarin 6); A_st_ and A_u_ represent the absorbance of the standard and sample at the excited wavelength, respectively; S_st_ and S_u_ are the integrated emission band areas of the standard and sample, respectively; n_Dst_ and n_Du_ are the solvent refractive index of the standard and sample; and subscripts u and s refer to the unknown and standard, respectively.

The effect of the metal cations upon the fluorescence intensity was examined by adding a few µL of stock solution (c = 10^−2^ mol L^−1^) of the metal cations to a known volume of the monomer solution (3 mL). The addition was limited to 0.08 mL, so that dilution remains insignificant [40]. For all absorption and fluorescence measurements, the dye concentration in solutions was 1 × 10^−5^ M. FTIR spectra were recorded on a Bruker IFS-113v spectrometer, Bruker, Karlsruhe, Germany, by the KBr pellet technique at a 2 cm^−1^ resolution. The NMR spectra were obtained on a Bruker DRX-250 spectrometer, Bruker, Karlsruhe, Germany, operating at 250.13 and 62.90 MHz for ^1^H and ^13^C, respectively, using a dual 5 mm probe head. DMSO-d_6_ and tetramethylsilane were used as a solvent and an internal standard, respectively. Thin layer chromatographic (TLC) analysis was performed on silica gel plates (Fluka F_60_ 254 20 × 20; 0.2 mm) using the solvent system n-heptane/acetone (1:1) as an eluent, if not mentioned otherwise. The following salts were used as a source for metal cations: MgCl_2_·6H_2_O, Pb(NO_3_)_2_, SrCl_2_, FeCl_3_·6H_2_O, AgNO_3_, CuSO_4_·5H_2_O, Co(NO_3_)_2_·6H_2_O, BaCl_2_·2H_2_O, and SnCl_2_·2H_2_O (Sigma-Aldrich, Taufkirchen, Germany).

### 2.1. Synthesis of N-(2-(6-((2-(dimethylamino)ethyl)amino)-1,3-dioxo-1H-benzo[de]isoquinolin-2(3H)-yl)ethyl)acetamide ***M1***

A mixture of **M0** (0.002 mol) and *N*,*N*-dimethylethylendiamine (0.0025 mol) in 5 mL of DMF was stirred at room temperature for 24 h. After that, water (100 mL) was added to the solution, the precipitate was filtered off, washed several times with water, and then air-dried at 40 °C. Yield: 89%.

FT-IR (cm^−1^): 3323, 2956, 2931, 2872, 1683, 1629, 1558, 1350, 1105, 1062, 777, 758;

^1^H-NMR (250.13 MHz, DMSO-d_6_): 8.46–8.45 (d, *J* = 7.27 Hz, 1H), 8.42 (d, *J* = 8.17 Hz, 1H), 7.94 (t, *J* = 7.52 Hz, 1H), 7.80 (t, *J* = 7.82 Hz, 1H), 7.33 (d, *J* = 7.69 Hz, 1H), 4.09 (t, *J* = 7.89 Hz, 2H), 3.34 (q, 6H), 3.27 (s, 4H), 2.30 (s, 3H), 1.68 (s, 3H);

^13^C-NMR (62.9 MHz, DMSO-d_6_): d 169.8, 164.4, 163.5, 151.0, 134.7, 131.0, 130.1, 128.8, 124.6, 122.4, 120.5, 111.3, 108.1, 104.2, 45.0, 37.1, 23.0, 21.6, 12.0;

Anal.: Calcd. (%) for C_20_H_24_N_4_O_3_ (368.1): C 65.20; H 6.57; N 15.21. Found: C 65.13; H 6.50; N 15.28.

### 2.2. Synthesis of N-(2-(6-(4-methylpiperazin-1-yl)-1,3-dioxo-1H-benzo[de]isoquinolin-2(3H)-yl)ethyl)acetamide (***M2***)

The synthesis was carried out according to the procedure described for **M1** using *N*-methylpiperazine as reactant. Yield: 86%.

FT-IR (cm^−1^): 3292, 2937, 2792, 1685, 1645, 1359, 1346, 1232, 1180, 1008, 786, 761;

^1^H-NMR (250.13 MHz, DMSO-d_6_): 8.89 (d, *J* = 7.07 Hz, 1H), 8.68 (d, *J* = 8.17 Hz, 1H), 8.20 (t, *J* = 7.55 Hz, 1H), 7.93 (t, *J* = 7.47 Hz, 1H), 7.87 (t, *J* = 7.21 Hz, 1H), 7.05 (d, *J* = 8.03 Hz, 1H), 4.33 (t, *J* = 7.20 Hz, 2H), 3.75 (t, *J* = 7.44 Hz, 2H), 2.75 (s, 6H), 1.97 (s, 3H);

^13^C-NMR (62.9 MHz, DMSO-d_6_), d: 169.8, 164.3, 163.8, 156.0, 132.5, 131.0, 130.8, 129.8, 126.5, 125.8, 123.3, 116.3, 115.5, 55.1, 53.0, 46.2, 36.9, 23.0;

Anal.: Calcd. (%) for C_21_H_24_N_4_O_3_ (380.3): C 66.30; H 6.36; N 14.73. Found C 66.39; H 6.31; N 14.79.

### 2.3. Synthesis of N-(2-(1,3-dioxo-6-(propylamino)-1H-benzo[de]isoquinolin-2(3H)-yl)ethyl)acetamide (***M3***)

The synthesis was carried out according to the procedure described for **M1** using n-propylamine as reactant. Yield: 94%.

FT-IR (cm^−1^): 3332, 2995, 2777, 1678, 1654, 1361, 1188, 1109, 1062, 1022, 779, 754;

^1^H-NMR (250.13 MHz, DMSO-d_6_): 8.69 (d, *J* = 7.47 Hz, 1H), 8.41 (d, *J* = 8.33 Hz, 1H), 8.23 (d, *J* = 7.47 Hz, 1H), 7.94 (t, *J* = 7.11 Hz, 1H), 7.65 (t, *J* = 7.59 Hz, 1H), 7.61 (t, *J* = 7.84 Hz, 1H), 4.07 (t, *J* = 7.42 Hz, 2H), 3.09 (m, 2H), 1.73–1.64 (m, 4H), 0.98 (t, *J* = 8.01 Hz, 3H);

^13^C-NMR (62.9 MHz, DMSO-d_6_), d: 169.8, 164.4, 163.5, 150.9, 134.7, 131.1, 130.0, 128.7, 124.9, 122.4, 120.5, 108.3, 104.2, 57.3, 45.7, 41.3, 37.0, 23.0;

Anal.: Calcd. (%) for C_19_H_21_N_3_O_3_ (339.2): C 67.20; H 6.24; N 12.38. Found: C 67.31; H 6.28; N 12.30.

### 2.4. Computational Details

The geometry optimization and photophysical properties of compounds were modeled with G16 software package [41]. The optimization of the ground and excited state geometry for **M1** and **M2** was performed within DFT [42] and TDDFT [43,44,45] formalisms, respectively. The theoretical computations have been carried out using B3LYP [46,47], PBE0 [48,49], and M06-2X [50] functionals in conjunction with 6-31+G(d,p) [51] and 6-311++G(d,p) [52] basis sets. Vibrational frequencies were evaluated for each structure at the same method/basis set to verify that the structures are indeed a minimum of the potential energy surface, and no imaginary frequency was found. The absorption wavelengths were determined by TDDFT calculations of vertical excitations. To simulate the fluorescence, the optimization of the excited state, corresponding to HOMO→LUMO transition was performed at TDDFT. Solvent effects were examined at each step by means of PCM formalism [53,54]. All the computations were performed in acetonitrile to reproduce the experimental conditions.

## 3. Results

### 3.1. Synthesis of Compounds ***M1***–***M3***

4-((2-dimethylaminoethyl)amino)-*N*-(2-acetamidoethyl)-1,8-naphthalimide (**M1**), 4-(4-methylpiperazin-1-yl)-*N*-(2-acetamidoethyl)-1,8-naphthalimide (**M2**), and 4-(n-propylamino)-*N*-(2-acetamidoethyl)-1,8-naphthalimide (**M3**) were obtained in good yields by the nucleophilic substitution of 4-nitro-*N*-(2-acetamidoethyl)-1,8-naphthalimide (**M0**) using primary amines in DMF solution at room temperature for 24 h (Scheme 1) according the method described previously [55]. The choice of the substituent attached to the imide nitrogen atom of the chromophore system is due to the fact that the new compounds can be considered as a structural element of modified with 1,8-naphthalimides polyamidoamine (PAMAM) dendrimers from different generations [5,35,36].

### 3.2. Spectral Properties

The spectral characteristics of newly synthesized compounds **M1**–**M3** have been evaluated in various organic solvents, and the results are summarized in Table 1: absorption (λ*_abs_*) and fluorescence (λ*_flu_*) maxima, Stokes shifts (Δ*ν_St_*), molar absorptivity (ε), and fluorescence quantum yield (Φ_st_).

In all the studied organic solvents, the compounds have an intense yellow color. Their absorption maxima depend on the type of substituents at the C-4 position and are respectively at 426–436 nm for **M1**, 394–406 nm for **M2**, and 427–444 nm for **M3**. This values are typical for 1,8-naphthalimides with intramolecular charge transfer (CT) [56]. The CT transitions are manifested in a substantial bathochromic shift compared to nitro-substituted **M0** (Figure 1) and the dependence of the maxima positions by the solvent polarity and especially their proton-donation ability. Similar behavior has been observed in the case of dendrimers [5,35,36].

From Figure 1, it can be seen that the absorption maximum of compound **M2** is hypsochromically shifted by about 40 nm compared to those compounds **M1** and **M3**, which have a mono-alkylamino group as a substituent on the C-4 group. This large difference can be explained by the interaction of the hydrogen atom at position C-5 and the substituents at C-4, which disrupts the planarity and respectively leads to a lower degree of conjugation [57]. All three compounds emit yellow-green fluorescence with maxima at the range 502–534 nm for **M1** and **M3**. The respective maxima of **M2** are slightly hypsochromically shifted compared to those of **M1** and **M3** (Table 1).

The Stokes shift is an important parameter indicating the difference between the structure and functional properties of the 1,8-naphtalimide fluorophores in their exited (S_1_) and ground (S_0_) states. It can be calculated by Equation (2).
(2)$$\Delta \nu_{St}=\nu_{abs}-\nu_{flu}=\left(\frac{1}{\lambda_{abs}}-\frac{1}{\lambda_{flu}}\right)10^{-7}{\mathrm{cm}}^{-1}$$

The obtained results have shown that the polarity of the organic solvents has a small effect on the Stokes shift values, while the type of substituent at C-4 shows a more significant effect. The presence of a cyclic amine leads to a larger Stokes shift.

An important characteristic of organic fluorophores, which are used as a signal unit in the design of fluorescent sensors, is their quantum efficiency. Quantum fluorescence yield Φ_F_ has been used for quantitative characterization. In polar solvents, compounds **M1** and **M2** have a low quantum yield, which is enhanced in non-polar media. This effect can be explained by the photoinduced electron transfer (PET) [2,3,4]. In the case of **M1**, the dependence of the quantum yield on the polarity of the solvents is better expressed in comparison to **M2**. In compound **M3**, where PET is not possible, no such dependence has been observed. These results indicate that compound **M1** and **M2** are good candidates for fluorescent sensors for metal ions and protons based on PET.

The character of the transitions and the influence of the substituents at position C-4 on the spectral properties were also studied with DFT/TDDFT computations. Various conformers for **M1** and **M2** have been investigated at the B3LYP/6-31+G(d,p) and M062X/6-31+G(d,p) level of theory and later, only the most stable conformers of **M1** and **M2** were optimized at B3LYP/6-311++G(d,p) (Figure 2). Below, only the properties of the most stable conformers will be discussed.

Both compounds consist of two parts, the naphthalimide (NI) moiety and amino substituent at position C-4. The NI part represents the fluorophore, and the amino substituent is the chelating unit. The geometry optimization of **M1** and **M2** in the ground state showed indeed that there is a steric repulsion between the H atom at position 5 and the substituents at the nitrogen (numbering is given in Scheme 2). The dihedral angle τ(C1C2NH) between the naphthalimide plane and H from amino substituent in **M1** is 14.5°, while the same angle τ(C1C2NC3) in **M2** is 63.9°. In **M1**, the plane structure is maintained, thus favoring the interaction between the NI moiety and the lone pair of N atoms in the substituent, suggesting that an efficient p–p conjugation can operate between the donor and acceptor units in **M1**. The C–N bond in **M1** is 1.353 Å and 1.396 Å in **M2** at B3LYP/6-311++G(d,p) level of theory.

Absorption and emission spectra in acetonitrile were also theoretically estimated using TDB3LYP, TDPBE0, and TDM062X levels of theory. The predicted absorption wavelengths of the lowest electronic transitions and oscillator strength for **M1** and **M2** using different functionals are listed in Table 2. Benchmark calculations on the absorption properties of various systems have demonstrated that the expected accuracy of TDDFT is between 0.2 and 0.3 eV [59,60]. Note that such accuracy can only be reached with a physically reliable description of transitions where the surroundings of the fluorophores and the performance of the method strongly depend on the system and the chosen functional. Vertical excitation and de-excitation energies without any state-specific correction are reported herein.

The main difference in the optimized structures between the ground S_0_ and excited S_1_ states concern the C–N bond lengths and the angle τ(C_1_C_2_NC_3_). For **M1**, the plane molecular structure is preserved also in the first excited state S_1_. However, the geometry of **M2** differs more dramatically between the ground and excited state—the dihedral angle τ(C_1_C_2_NC_3_) increased to 88.0° (from 63.9°). In addition, the C–N bond was elongated from 1.396 Å in S_0_ to 1.436 Å in S_1_ at the B3LYP/6-311++G(d,p) level of theory.

The trends of changes in electron densities can be illustrated through a molecular orbitals shape analysis. Figure 3 shows the ground state orbital energy levels of the highest occupied molecular orbital and the lowest unoccupied molecular orbital and the energy gap for **M1** and **M2** in acetonitrile.

The HOMO of **M2** is delocalized over the naphthalimide moiety and the donor (*N*-methylpiperazine), while LUMO covers the acceptor part. The orbitals’ shape indicates that an effective intramolecular charge transfer occurs from the donor group toward the imide moiety. Hence, HOMO→LUMO transition is classified as an internal charge transfer (ICT) transition with contribution from *N*-methylpiperazine substituent. The small sensitivity toward solvation effect can be explained with the small change of dipole moment between the ground and excited states. The combination of a functional and basis set used for the spectral calculations proved to be reliable for the studied 1,8-naphthalimides. The ICT transition is simulated with good accuracy from the theoretical calculations. The B3LYP functional looks even “too good” for calculating the CT transition in **M1** (Table 2). The calculated fluorescence maxima in acetonitrile for the **M1** at B3LYP/6-311++G(d,p) level is 2.43 eV (510 nm), and the experimental value is 2.44 eV (509 nm). In the case of **M2**, it overrated the wavelength of the HOMO→LUMO transition (Table 2). The next transition S_2_ (HOMO-1→LUMO) correlates well with the experimental results. The ICT transition in **M2** using M062X functional predicts that the S_1_ excited state consists of HOMO→LUMO (93%) and HOMO-1→LUMO (4%) transitions.

The excited state geometry for **M2** was optimized at B3LYP/6-311++G(d,p) level and predicts the fluorescence maxima in acetonitrile at 2.56 eV (485 nm and oscillator strength f = 0.16), which is in an acceptable agreement with the experimental value (2.36 eV). Briggs and Besley stated that consideration of the MO only in the ground state is not sufficient to reliably predict the photoinduced electron transfer [61]. The energies and shape representation of the frontier molecular orbitals for optimized excited state S_1_ geometry of **M2** obtained from TDB3LYP/6-311++G(d,p) computations are given in Figure 3.

As can be seen from Figure 4, in **M2** the electron density of HOMO is distributed over the N-methylpiperazine moiety—the receptor, whereas HOMO-1 is localized also on the naphthalimide unit—the fluorophore. The electron density of LUMO is centered on the 1,8-naphthalimide part—the fluorophore. The lone pair orbitals of the amino groups in the receptor are of higher energies than those of the HOMO of fluorophore, which is a precondition for a typical reductive PET process. So, when excitation occurs, an electron would be transferred from the receptor to the fluorophore, which will lead to the quenching of **M2** emission.

### 3.3. Detection Ability of Metal Cations of ***M1*** and ***M2***

The potential of **M1** and **M2** for detection of metal ions has been evaluated by titration with metal ions (Ag^+^, Mg^2+^, Sn^2+^, Pb^2+^, Sr^2+^, Cu^2+^, Co^2+^, Ba^2+^ and Fe^3+^,) in acetonitrile solution. As can be seen from the data in Table 2, the fluorescence intensities of **M1** and **M2** in acetonitrile are very low due to the PET effect. Also, the metal salts and their complexes are soluble in it. All of this makes acetonitrile a good solvent in the study of the sensor activity of the **M1** and **M2** ligands. After the addition of metal ions, the fluorescence of **M1** and **M2** is enhanced due to the quenching of the PET. This effect can be quantified using the enhancement of the fluorescence intensity (FE = I/Io), which is the ratio of the maximum fluorescence intensity (I, after the addition of metal ions) and the initial fluorescence intensity (Io, ligand before adding metal ions). Figure 5 shows the results obtained for FE for the compounds **M1** and **M2**. FE has been recorded only when titrating with metal ions of groups IVA and VIIIB, which shows a good ability of the compounds to detect such elements. The highest values of FE for both compounds have been obtained for Fe(III) ions: FF = 32.5 for **M1** and FF = 243.6 for **M2**. Some of the other metal ions, Sn^2+^, Co^2+^, Pb^2+^ when using **M1** and Sn^2+^, Co^2+^ for **M2** have similar values.

The structure of the complex between **M2** and Fe^3+^ ion was fully optimized in gas phase at the B3LYP/6-31G(d) level of theory (Figure 6). Vibrational analysis was performed at the same level of theory. No imaginary frequency was found for the optimized structure, indicating a local minimum of the potential energy surface. The metal cation binds to both nitrogen atoms in a bidentate manner. The metal coordination number is 6 in agreement with the experimental observations. The other metal ions form the same complexes. When using **M1** as a ligand, the complexes are formed with the two nitrogen atoms of the receptor fragment of the 1,8-naphthalimide structure (-**NHCH_2_CH_2_N(CH_3_)_2_**). These results are in very good accordance with our other research studies [5,24,27,32].

A representative example from the titrations of **M1** has shown in Figure 7. In the presence of Sn^2+^ ions, its fluorescence intensity has been dramatically increased with a small change in the position of the fluorescence maxima (Δλ_F_ = 6 nm).

The titration profile shows a linear increase of the signal as the concentration of the analyte increases, until the latter reaches the concentration of the compound **M1**, followed by a plateau, indicating a 1:1 metal-to-ligand ratio for the complex formation (Figure 7A). It should be noted that excellent signal-to-noise ratios are observed even for cation concentrations around and below 1 × 10^−6^ M. The dependence of the fluorescence intensity of **M1** on the concentration of Fe (III) ions in the concentration range 0÷1 × 10^−5^ mol L^−1^ shows that very good linear dependence has been obtained R = 0. (Figure 7B) The limit of detection (LOD) = 5.77 × 10^−7^ mol L^−1^ and the limit of quantitation (LOQ) = 1.91 × 10^−6^ mol L^−1^ have been calculated on the basis of linear regression [24]. The results obtained for other metal ions have been in the same order. This indicates that **M1** and **M2** can be used for the detection of these metal ions in the environment at the ppm concentration range.

### 3.4. Influence of pH on the Fluorescence Intensity of the ***M1*** and ***M2***

In a water–ethanol mixture (1:4 *v*/*v*), the fluorescence emission of **M1** and **M2** has been studied by varying the pH values in the range pH = 3 ÷ 11 (Figure 8). The results show the high pH sensitivity of both compounds. In the case of **M1**, with increasing pH values up to 8, a plateau of the fluorescent intensity has been observed, after which it sharply decreases and at pH > 9, it maintains its low values. A similar dependence has been observed for compound **M2**, but the decrease in fluorescence intensity was smoother and started at pH = 5.5.

The quantitative pH dependence of the fluorescence intensity of compounds **M1** and **M2** was analyzed using Equation (3), and their pKa values have been calculated.
(3)pH−pKa=log[(Imax−I)(I−Imin)]

For compound **M1**, pKa was 7.79, while **M2** has a slightly higher value pKa = 8.27. As can be seen from the figure, the fluorescent intensity is significantly higher in an acidic environment where it is possible to protonate the tertiary amino group from the receptor fragment and block the PET process. The observed pH value of the fluorescence intensity is more pronounced in the case of **M2**, where the fluorescence intensity increases by a factor of more than 50 upon the transition from alkaline to acidic medium, while in the case of **M1**, the increase is 18 times. A hypsochromic shift of the fluorescence maxima of 13 nm for **M1** and 20 nm for **M2** has also been observed. These results indicate that compounds **M1** and **M2** could be used as pH sensors.

## 4. Conclusions

The photophysical characteristics of three new 1,8-naphthalimide compounds (**M1**, **M2**, **M3**) have been evaluated in organic solvents of different polarity. The influence of the substituent at C-4 position has been investigated, and it was found that the chemical structure has a significant effect on these values. For compounds **M1** and **M2** containing tertiary amino groups connected to the chromophore system via an ethylene spacer, the fluorescence intensity depends strongly on the polarity of the medium. For compound **M3**, this dependence is negligible. This indicates that the tested compounds **M1** and **M2** can be used in the design of PET-based sensor systems. By using DFT and TDDFT computations, the sensor mechanism was rationalized, and the reductive PET mechanism was confirmed. The effects of different metal ions (Ag^+^, Ba^2+^, Cu^2+^, Co^2+^, Mg^2+^, Pb^2+^, Sr^2+^, Fe^3+^, and Sn^2+^) on the fluorescence intensity of the two compounds have been investigated. It has been shown that in the presence of Ag^+^, Pb^2+^, Sn^2+^, Co^2+^, and Fe^3+^ the fluorescence intensity is enhanced, and at compound **M2**, this effect has been better expressed. Both compounds exhibited pH dependence on the fluorescence intensity, which is low in an alkaline medium and enhanced in acidic medium.

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
