# Peer review of "Synthesis, Photophysical Characterization, and Sensor Activity of New 1,8-Naphthalimide Derivatives"

_sensors, 2020, doi:10.3390/s20143892_

Round 1

Reviewer 1 Report

The paper entitled: Synthesis, photophysical characterisation and sensor activity of new 1,8-naphthalimide derivatives, by: S. Yordanova-Tomova, D. Cheshmedzhieva, S. Stoyanov, T. Dudev, and I. Grabchev, deals with the preparation of a few naphthalimide derivatives and their physicochemical description, as well as some experiments of fluorescent changes in the presence of several common cations in acetonitrile. The compounds appear to be new, albeit they are closely related to several related naphthalimides previously published by several authors. There is no clear explanation about the reasons to select those compounds presented in the paper. The immediate conclusion is that two of the naphthalimides are sensitive to acidic metal cations as well as to the pH of the solution. A representative example of fluorescent titration and titration plot for one of the compounds with one metal cation and LOD and LOQ for a different cation, therefore the interested reader of the paper will probably be lost when looking for the applications of the results in the paper. I think that the paper may be interesting but the results are presented in a very preliminary state. Nothing is said about the interference or effect of different common anions in the fluorescent changes of the compounds, nothing about the influence of water in the sensing action of the compounds for representative important cations to be detected. Detection of iron, cobalt or tin cations in acetonitrile is not very useful for environmental purposes but the detection of lead cation, which is selective for one of the compounds, is probably the most interesting finding of the paper but nothing is said about the fact. I think that before publication the authors should give a complete description of the sensing capabilities of the naphthalimides presented in the paper and a plausible practical application to a representative environmentally important problem, for example the selective detection of lead cation in water or solvent-water samples by the crossed titration with the two naphthalimides to discard interference from common cations such as iron or tin cations in the same solution. With that application I think that the paper could be publishable. A supporting information with NMR spectra and, eventually, high resolution mass spectra, will help to the publication of the paper, including also the fluorescent titration and titration plots of the cations as well as interferent compounds under study.

Author Response

The paper entitled: Synthesis, photophysical characterisation and sensor activity of new 1,8-naphthalimide derivatives, by: S. Yordanova-Tomova, D. Cheshmedzhieva, S. Stoyanov, T. Dudev, and I. Grabchev, deals with the preparation of a few naphthalimide derivatives and their physicochemical description, as well as some experiments of fluorescent changes in the presence of several common cations in acetonitrile.

The compounds appear to be new, albeit they are closely related to several related naphthalimides previously published by several authors.

All three compound are novel.

There is no clear explanation about the reasons to select those compounds presented in the paper. The immediate conclusion is that two of the naphthalimides are sensitive to acidic metal cations as well as to the pH of the solution.

The design of these compounds was aimed at obtaining compounds  which have at the C-4 position a receptor fragment (M1 and M2) or a substituent without receptor properties (M3). The substituent on the imide nitrogen atom is N-acethylethylenedianine.These compounds can be considered as basic structural elements of  modified PAMAM dendrimers with 1,8-naphthalimide which we have been doing in our laboratory for years. This will allow us to evaluate the influence of the structure of 1,8-naphthalimides after their binding to the dendrimer periphery on their photophysical characteristics and sensory activity and  on the other hand, it will allow us to evaluate the multivalent effect of dendrimers

A representative example of fluorescent titration and titration plot for one of the compounds with one metal cation and LOD and LOQ for a different cation, therefore the interested reader of the paper will probably be lost when looking for the applications of the results in the paper.

Yes, this is just one example of the order in which these values are under the conditions studied

I think that the paper may be interesting but the results are presented in a very preliminary state. Nothing is said about the interference or effect of different common anions in the fluorescent changes of the compounds, nothing about the influence of water in the sensing action of the compounds for representative important cations to be detected. Detection of iron, cobalt or tin cations in acetonitrile is not very useful for environmental purposes but the detection of lead cation, which is selective for one of the compounds, is probably the most interesting finding of the paper but nothing is said about the fact.

As mentioned above, the purpose of the synthesis of those compounds is, on the one hand, to characterize their spectral properties and, on the other hand, to test their detection ability for metal ions or protons. We have used traditional metal ions, and the choice of solutions such as acetonitrile is dictated by the fact that the synthesized dendrimers have so far been tested in this solution or in DMF. Therefore, in this case we did not look for the influence of water.

I think that before publication the authors should give a complete description of the sensing capabilities of the naphthalimides presented in the paper and a plausible practical application to a representative environmentally important problem, for example the selective detection of lead cation in water or solvent-water samples by the crossed titration with the two naphthalimides to discard interference from common cations such as iron or tin cations in the same solution. With that application I think that the paper could be publishable.

The real practical application of this manuscript is to characterize the functional properties of these compounds, which will serve for a more complete and detailed characterization of the dendrimers, which have been published by our group.

A supporting information with NMR spectra and, eventually, high resolution mass spectra, will help to the publication of the paper, including also the fluorescent titration and titration plots of the cations as well as interferent compounds under study.

Reviewer 2 Report

This paper is on the synthesis of three novel naphthalimide derivatives, M1-M3, and the performance of M1 and M2 as PET-type molecular sensors for metal ions. My main problem with this paper is that the synthetic route described and the results of quantum chemical calculations on the PET mechanism may be more interesting than the analytical section, the latter representing a limited value in terms of novelty. The construction of fluorescent chemosensors for the detection of metal ions in acetonitrile was a popular subject 15-20 years ago. Since then a variety of more sophisticated fluorescent sensors have been synthesized for the detection of complex biomolecules like amino acids or nucleotides, which function in aqueous media.

Further comments

  • the synthetic route for M1-M3 is rather special: one would expect a synthesis from a 4-Br and not from 4-NO2 naphtalimide M0. No reference is presented on the direct NO2 -> NHR substitution reaction.
  • Scheme 1: It is ambiguous how the A substituents are attached to the naphthalimide ring system
  • what does ‘state specific corrections’ (row 202) mean?
  • At least a hypothetical structure for a metal complex should be presented

Author Response

This paper is on the synthesis of three novel naphthalimide derivatives, M1-M3, and the performance of M1 and M2 as PET-type molecular sensors for metal ions. My main problem with this paper is that the synthetic route described and the results of quantum chemical calculations on the PET mechanism may be more interesting than the analytical section, the latter representing a limited value in terms of novelty. The construction of fluorescent chemosensors for the detection of metal ions in acetonitrile was a popular subject 15-20 years ago. Since then a variety of more sophisticated fluorescent sensors have been synthesized for the detection of complex biomolecules like amino acids or nucleotides, which function in aqueous media.

Further comments

  • the synthetic route for M1-M3 is rather special: one would expect a synthesis from a 4-Br and not from 4-NO2 naphtalimide M0. No reference is presented on the direct NO2 -> NHR substitution reaction.

We used 4-nitro-1,8-naphthalimide because the nucleophilic substitution of nitro group with the amino one, unlike its bromine analogue, takes place at room temperature. Рespective references  were cited in the text.

  • Scheme 1: It is ambiguous how the A substituents are attached to the naphthalimide ring system

Corrected

  • what does ‘state specific corrections’ (row 202) mean?

This term concerns the computation of the solvent response. In a vertical transition the redistribution of electron density upon photon absorption is assumed to be faster than the solvent nuclei redistribution. This is a “nonequilibrium” solvation effect. Vertical excitation energies are calculated using nonequilibrium solvation (with the static solvation from the ground state). State specific (SS) solvation calculations (‘‘state specific correction’) account for the redistribution of the solvent. State specific (SS) solvation calculations are more demanding from computational standpoint and at the same time it was shown that for a series of substituted 1,8-naphthalimides do not provide better conformity with the experiment (D. Cheshmedzhieva, P. Ivanova, S. Stoyanov, D. Tasheva, I. Ivanov, S. Ilieva “Absorption and fluorescence properties of novel 1,8-naphtalimide hydrazones for enzyme activity localization” Phys. Chem. Chem. Phys. 13, 2011, 18530-18538.)

  • At least a hypothetical structure for a metal complex should be presented

Done in the text

Round 2

Reviewer 1 Report

The paper entitled: Synthesis, photophysical characterisation and sensor activity of new 1,8-naphthalimide derivatives, by: S. Yordanova-Tomova, D. Cheshmedzhieva, S. Stoyanov, T. Dudev, and I. Grabchev, deals with the preparation of a few naphthalimide derivatives and their physicochemical description, as well as some experiments of fluorescent changes in the presence of several common cations in acetonitrile. Albeit the compounds are new, they are closely related to several related naphthalimides previously published by several authors. Two of the naphthalimides are sensitive to acidic metal cations as well as to the pH of the solution and a representative example of fluorescent titration and titration plot for one of the compounds with one metal cation and LOD and LOQ for a different cation are presented, therefore the applications of the results in the paper are not fully developed. The paperhas been corrected in the motivation and some details but the results are essentially the same and are therefore studied only in a very preliminary state. Nothing new is presented in the corrected paper therefore I think that it could only be publishable as a preliminary study of the synthesis of some fluorogenic probes and the interactions with metal cations on the way to sensor properties, which are not developed.

Author Response

We agree that these compounds are similar to those already published by other authors, but only in the part of the 1,8-naphthalimide structure where the receptor fragments are (at C-4 position). We have used two receptor fragments that are widely used in the design of 1,8-naphthalimide sensors. Like and other authors, we have used them in our previous research. Probably these receptors will be used by other authors in the future at their research on the sensor systems. The novelty of our compounds is the introduction of the substituent at the imide nitrogen atom, which is a structural element of the interior part of PAMAM dendrimers. It was the aim of these studies to accumulate experimental material on how the binding of the sensory 1,8-naphthalimide fragment to the dendrimers affects the sensor capacities of the dendrimers, in the structure of which there are a large number of such signaling units. We have used similar experimental conditions as we used in the study of dendrimers modified with these 1,8-naphthalimides to make the comparison correct.

Reviewer 2 Report

My comments have been taken into account in the revised manuscript.

Author Response

Thanks to the reviewer!